# A global Youth Peacebuilding Beliefs Scale
Laura K. Taylor [1,7], Vivian Liu [1,7] ✉, Bethany Corbett[2], Juliana Valentina Duarte Valderrama[3], Léïla Eisner[4], Jeanine Grütter [5], Eran Halperin [6], Tabea Hässler [4], Claudia Pineda-Marin[3] & Ilana Ushomirsky [6]

Youth are often framed as victims or perpetrators of conflict. Yet, they also can disrupt conflict cycles as peacebuilders. Motivated by SDG 16 and UN Security Council Resolutions 2250, 2419, and 2535 — recognising and facilitating youth participation in fostering peace and social inclusion — we developed and validated a global Youth Peacebuilding Beliefs Scale (YPBS), a novel measure of different types of peacebuilding across levels of the social ecology (i.e., microsystem, mesosystem, and macrosystem). We used a sequential mixed-methods, cross-cultural design with adolescents (ages 14-17) and young adults (ages 18-26) across two studies (Study 1: Focus groups, N = 199, Northern Ireland n = 78, Colombia n = 60, Israel n = 41, Switzerland n = 20; Study 2: Survey, N = 2771, Northern Ireland n = 514, Colombia n = 806, Israel n = 833, Switzerland n = 618) across four diverse cases to explore youth's contributions along the peace continuum from active conflict to stable democracy. The YPBS provided an empirical test of the Developmental Peacebuilding Model and can be used by policymakers and researchers to support youth-driven quality peace.

2023 marked the highest number of conflicts since World War II, driving a record 117 million people from their homes[1]. With nearly 60% of conflicts reigniting within 10 years[2,3], global action is urgently needed to break these cycles of violence. Advancing the United Nations (UN) Sustainable Development Goal (SDG) 16 on Peace, Justice and Strong Institutions, we examined peacebuilding through collective action. Motivated by the UN Security Council Resolutions (2250[4], 2419[5], 2535[6]) that recognise and facilitate young people's participation in preventing conflict, sustaining peace, and fostering inclusion, we developed and validated a global Youth Peacebuilding Beliefs Scale (YPBS). Consistent with the UN's multi-level approach to youth peacebuilding, the YPBS can help measure how youth interrupt the cycles of violence as agents of peace around the world.

Our approach addressed two gaps in the literature. First, challenging the depiction of young people as victims or perpetrators[7], we investigated how youth—one-third of the world's population—can be peacebuilders[8–10]. Across political science, sociology, economics, and psychology, youth typically remain an understudied demographic whose *constructive* agency is overlooked[11,12]. Yet, nonviolent civic engagement (e.g., voting, peaceful protests, volunteering) is key for building peace and youth are particularly effective in these types of collective action[13–18]. The constructive impact of youth involvement has been witnessed in global movements such as Black Lives Matter[19,20] and Fridays for Future[21]. Moreover, strong localised case

studies[11,22–26] have begun to document youth at the forefront of peacebuilding in conflict settings[9,10,27–30].

Second, decades of research has studied peacebuilding in conflict settings[31–34]; instead, we adopted the peace continuum from current violent conflict to no recent history of violent conflict[35,36]. Under this framework, peacebuilding can address different phases and intensities of direct violence[37–39] (i.e., violent or armed conflict), structural violence[40] (i.e., systematic oppression or inequality), and cultural violence (i.e., norms or beliefs used to legitimise direct or structural violence[41]). From this perspective, peacebuilding can promote quality peace even in non-conflict contexts, such as stable democracies[36,42].

We addressed these two gaps through a cross-cultural study across the peace continuum to develop and validate a Youth Peacebuilding Beliefs Scale (YPBS). Our sequential mixed-methods approach was rooted in the Developmental Peacebuilding Model (DPM[9]), which integrates foundational frameworks from developmental psychology[43] and peace science[31]. The DPM outlines the predictors and impact of youth peacebuilding across three *types* of peacebuilding – microsystem, mesosystem, and macrosystem[12] – or three levels of social change. Microsystem peacebuilding consists of interpersonal interactions for relational change (e.g., prosocial behaviour targeting an individual, such as helping or sharing[44–49]); mesosystem peacebuilding involves actions targeting structural change (e.g.

[1]University College Dublin, Dublin, Ireland. [2]Ulster University, Belfast, UK. [3]Konrad Lorenz University, Bogotá, Colombia. [4]University of Zurich, Zürich, Switzerland. [5]Ludwig Maximilian University of Munich, Munich, Germany. [6]Hebrew University of Jerusalem, Jerusalem, Israel. [7]These authors contributed equally: Laura K. Taylor, Vivian Liu. ✉e-mail: vivian.liu@ucd.ie

collective action to change laws to help disadvantaged social groups[50,51]); and macrosystem peacebuilding advances cultural change (e.g. changing overarching systems[52] – e.g., voting[53] or norms[54]).

The DPM argues that peacebuilding is sustained prosocial action vertically integrated across different levels[53]. Frequently, a given study or scale only addresses a single type of peacebuilding: For example, one-on-one interactions with conflict rivals[55,56] (e.g., microsystem or relational change), or collective action to challenge a regime[16–18,57–59] (e.g. mesosystem, or structural change). Macrosystem peacebuilding (i.e. cultural change) in youth is largely under-researched[60]. Thus, the YPBS is innovative in its approach to measure the DPM's three types of peacebuilding across the peace continuum.

We offer another significant advancement by empirically testing the YPBS with two age groups. Many studies often focus on one age group (i.e., adolescents or young adults). The DPM enabled us to examine how different types of peacebuilding may co-occur across development to understand youth peacebuilding as more than the sum of its parts.

We strategically selected four cases to investigate the robustness of the YPBS across the peace continuum. We started with Northern Ireland, a post-conflict society where youth today are a full generation away from the 1998 peace agreement that ended The Troubles. As one of the most widely studied conflicts[61], the peace process in Northern Ireland is commonly referenced as success case[62,63] and served as the baseline comparison in our research. Colombia is only recently post-conflict; youth today are half a generation since the 2016 peace agreement ending the 50-year civil war, though both regional direct violence (i.e., remaining paramilitary groups) and structural violence (i.e., inequality) remain[64–66]. Israel has ongoing and high-intensity conflict; youth today have not seen the fruits of past attempts at peace[67,68]. Finally, Switzerland represented the other end of the spectrum with no recent history of violent conflict and one of the lowest levels of inequality globally[69]; young people are growing up amid higher quality peace. We recognise that work on youth peacebuilding has been conducted in these cases separately (Northern Ireland[53,70,71], Colombia[72–76], Israel[77–79], and Switzerland[47,80,81]), but the strategic selection of these cases enabled us to examine the YPBS across the peace continuum, establishing its validity as a global scale. Within each case, we also explored how youth peacebuilding may vary by subgroup (e.g., age group, gender, intergroup identity). Taken together, this approach teased apart universal and unique aspects[82] to understand and measure youth peacebuilding.

The development and validation of the YPBS followed four key phases[83] (Fig. 1), using a sequential exploratory mixed-methods design. First, during *item development (Phase 1)*, focus groups were conducted with youth to generate items from their perspectives on different types of peacebuilding across the social ecology; this approach established content validity with the target population[84,85]. Second, during *scale development (Phase 2)*, items were reduced using factor analysis[86]; we hypothesised a

3-factor structure reflecting the DPM, also balancing parsimony and sufficient items to capture macrosystem peacebuilding. Third, during *scale validation (Phase 3)*, we examined dimensionality (confirmatory factor analysis, measurement invariance), reliability (Cronbach's alpha, McDonald's omega), and validity (convergent and discriminant) across the four cases. Finally, we briefly explored differences *across* and *within-case (Phase 4)*. Together, this approach advanced the YPBS's key contributions: (1) highlighting youth's multi-level role in peacebuilding, consistent with UN recommendations, (2) understanding youth's beliefs across a global peace continuum.

## Methods
### Study 1: Qualitative focus groups
**Participants.** Adolescent and young adult participants ($N = 199$; 130 women, 69 men; ages 14–26) were recruited across four cases: Northern Ireland ($N = 78$; 46 adolescents, 32 young adults; 63 women, 14 men; Protestants and Catholics), Colombia ($N = 60$; 30 adolescents, 30 young adults; 38 women, 22 men, 1 non-binary), Israel ($N = 41$; 19 adolescents, 22 young adults; 20 women, 21 men; Jewish Israeli) and Switzerland ($N = 20$; 11 adolescents, 9 young adults; 9 women, 8 men, 3 non-binary) from youth groups, high schools, universities, and online, relying on a mix of community-based recruitment and convenience sampling through existing connections with schools from our collaborators. No participants were excluded as they were all screened prior to participation to ensure they fulfilled the relevant age and nationality criteria (i.e., born or raised in the specific country). For a detailed demographic and recruitment methods, see Supplementary Tables S1, S2.

**Design and procedure.** Procedures were approved by the Ethics board at University College Dublin and the Ethics boards of the partner institutions: Konrad Lorenz University, The Hebrew University of Jerusalem, and the University of Zurich. All participants (and their guardians, if the participant was an adolescent) received an information sheet and provided consent prior to participating.

All focus groups were conducted by a trained research assistant in-person or on Zoom, lasting 60–90 minutes. Focus groups were recruited as part of a larger project on youth peacebuilding, following a semi-structured format. This study utilized a portion of the focus group discussion that asked about youth's experiences with different peacebuilding activities; for example, questions included "How much [the activity] is useful/helpful to peacebuilding in [case]? Why or why not?", "Can you share an example of you, a friend, or someone you know doing this? Tell us about how it looked or worked.", "Do you think many young people your age are, or have been, involved in/attended/participated in [the activity]?" The list of peacebuilding activities was compiled from previous peacebuilding and civic engagement research, selected to cover all levels of the DPM. For example, on the microsystem level, we asked about activities such as following the

**Fig. 1** | Graphical summary of key phases and methods to develop and validate a global Youth Peacebuilding Beliefs Scale (YPBS).

DEVELOPING AND VALIDATING THE YPBS

Study 1 Qualitative → Phase 1: Item Development — Focus groups to understand and generate an initial item pool

Study 2 Quantitative → Phase 2: Scale Development — Exploratory factor analysis for factor structure and parsimony

Phase 3: Scale Validation — Dimensionality, reliability, and validity across 4 cases

Phase 4: Across- and Within-Case (Exploratory) — Group differences by case, age, gender, and intergroup identity

https://doi.org/10.1038/s44271-025-00340-4                                                                                    **Article**

**Table 1 | Youth Peacebuilding Beliefs Scale (YPBS) by subscale**

| Subscale | # | Item |
|---|---|---|
| Macrosystem (Beliefs) | 1 | We should care less about group labels and get to know each other as individual people |
| | 2 | Learning about sectarianism/discrimination is important for peacebuilding |
| | 3 | Spending time with people from the other background reduces prejudices/negative stereotypes[a] |
| | 4 | The younger generation has the power to create a more peaceful future |
| | 5 | Conversations about peace need to focus more on the future and less on the past |
| Macrosystem (Voting/Politics) | 1 | Voting is an important way to express your beliefs |
| | 2 | Voting makes a difference in my country |
| | 3 | We need more young people in politics |
| Mesosystem | 1 | Protests are an effective way to build peace |
| | 2 | Protesting about social issues (e.g., LGBTQ, cost of living) can bring people together |
| | 3 | Protest art is a way to express your opinion[a] |
| Microsystem | 1 | Discussions can help raise awareness about peacebuilding |
| | 2 | It [discussions] helps me see a difference perspective |

[a]Item was omitted for the Israeli sample due to inapplicability to the cultural context.

news[87], doing volunteer work[87–89], and participating in discussions[90]. On the mesosystem level, we asked about protests[87,90–92], protest art[90], strikes[90,91], and boycotting[91,92]. On the macrosystem level, we asked about voting and getting involved in politics more broadly[87] (for the full list of activities, see Supplementary Note S3). In Northern Ireland, because intergroup contact with conflict rivals was possible and, occasionally common (unlike Colombia and Israel), case-specific questions about cross-community contact were also included[93–95].

Recordings were transcribed verbatim using an automatic transcription service and manually checked by a local research assistant. Final transcripts were pseudo-anonymised; Digital and human translation was conducted[96,97].

### Study 2: Quantitative survey

**Participants**. 977 adolescents and 1794 young adults from Northern Ireland ($N = 514$), Colombia ($N = 806$), Israel ($N = 833$), and Switzerland ($N = 618$) were recruited, following sampling and inclusion criteria preregistered on Open Science Framework: https://osf.io/52epw. These included (a) born or raised in the country and (b) within the specified age ranges.

**Design and procedure**. Participants completed a 20–30-minute-long online survey as part of a larger study on youth peacebuilding. Procedures were approved by the Ethics board at University College Dublin and preregistered: https://osf.io/52epw. All participants, and guardians of adolescents, received a detailed information sheet detailing the rationale, risk and benefits, and contact information for the study. All participants provided online consent, prior to completing the survey. In Colombia only, participants completed the survey with the help of a data collection assistant to accommodate technology accessibility. Participants were excluded if they were 1) not born or raised in the target site, 2) outside of the specified age range, or 3) failed the attention and quality checks embedded within the survey.

Results from Study 2 were used in both scale development (Phase 2) and scale validation (Phase 3). For purposes of developing and validating the YPBS, three additional measures were included for scale validation (Phase 3) across the three levels of the DPM. First, *societal responsibility* examined participants' broader thinking about society and their responsibility to participate in it, which corresponds to the macrosystem level. Second, *protest participation* is a direct behavioural measure of mesosystem peacebuilding, which corresponds to their attitudes (i.e., mesosystem YPBS subscale). Finally, *general prosocial behaviour* speaks to interpersonal instances of sharing or cooperative behaviour (i.e., microsystem), and has

been examined in past literature as an important antecedent to broader peacebuilding[98].

**Measures**. *Youth Peacebuilding Items*. Participants were asked to endorse 22 peacebuilding-related items on a scale of 1 (strongly disagree) to 5 (strongly agree). These items covered topics such as voting/politics (e.g., "Voting makes a difference in my country"), general peace beliefs (e.g., "The younger generation has the power to create a more peaceful future"), intergroup attitudes and knowledge (e.g., "Learning about sectarianism/discrimination is important for peacebuilding"), protests (e.g., "Protesting about social issues can bring people together), protest art (e.g., "Protest art is a way to express your opinion"), and discussions (e.g., "Discussions can help raise awareness about peacebuilding"); See Supplementary Table S7 for a full list of the original 22 items, and Table 1 for the 13 items retained in the final YPBS.

*Societal Responsibility* was assessed by three items "If you love [country], you should notice its problems and work to correct them", "I oppose some [country] policies because I care about my country and I want to improve it", and "Being concerned about national and local issues is an important responsibility for everybody" adapted from the Civic Responsibility Scale[99] and Political Efficacy Scale[100]. Higher scores indicated greater societal responsibility ($\alpha = 0.73$).

*Protest Participation* was assessed if they had participated in a protest, demonstration, or march. For analyses, responses were coded as a binary no (0) or yes (1) for protest participation.

*Prosocial Behaviour* was assessed with seven items from the prosocial behaviour subsection of the Child Behaviour Scale[101]; For example, "I am kind towards other people" and "I am cooperative with other people". Higher scores indicated greater interpersonal prosocial behaviour ($\alpha = 0.89$).

### Reporting summary

Further information on research design is available in the Nature Portfolio Reporting Summary linked to this article.

## Results

### Phase 1: Item development (Study 1)

The planned analyses generated items through focus group transcripts by three researchers (Author 1a, Author 1b, and a research assistant). First, all researchers read the translated transcripts and familiarized themselves with the data. Then, all researchers examined the transcripts line-by-line and compiled quotes about specific peacebuilding activities. Through further discussion, quotes were rephrased into scale items that could be asked on a

Likert scale of 1–5 (strongly agree to strongly disagree) (e.g. quote"[Voting] It's your chance to make a difference" was reworded into item "Voting makes a difference in my country"). This process was repeated for all types of peacebuilding activities. Finally, the research team reviewed the items generated and checked for distribution of items across the three levels of the DPM, specifically focusing on retaining a larger number of macrosystem level items to address the gap in the literature.

First, we found that across the four cases, youth noted the importance of peacebuilding activities across three levels of the DPM. For example, at the macrosystem level toward cultural change, youth discussed actions such as voting (e.g., "It's your chance to make a difference") and political participation (e.g., "Young people getting involved in voting and politics is important"), as well as changing norms around key social divides (e.g., "an unspoken rule", "Something very fundamental in the way our society is structured") and generational differences, ("We've now moved past [the Troubles]… whereas the older generation [hasn't]"). At the mesosystem level for structural change, youth mentioned joining protests or demonstrations (e.g., "Everybody knows somebody who's been on a protest and they are very much a way for us to voice our frustrations") and protest art (e.g., "It [protest art] gives young people a voice about an event that happened, that cannot be forgotten or denied"). Finally, at the microsystem level toward relational change, youth focused on discussing political issues with others (e.g., "talking about topics instead of pretending like they're not happening… even if it's just with your peers [you] can bring peace in some ways").

Second, some peacebuilding activities, however, were perceived differently across cases. For example, generational divides was especially important to Northern Ireland (e.g., "A lot of young people care less and less about the borders that have been put up between people in our community for the past"), and protest art emerged as a prominent form of peacebuilding for youth in Colombia (e.g., "[art] makes you reflect… it is giving a voice that many of us do not have"). In Israel, youth expressed difficulty about participation in general (e.g., "Powerlessness… we give up in advance and simply do not act"), in contrast to youth in Switzerland (e.g., "I think that [going on strike] is the easiest thing for young people.").

Third, across these different peacebuilding activities, a pool of 22 YPBS items was generated (see Supplementary Table S7 for the full list of items). These items were reviewed by our collaborators (Authors 2–8); with consideration of applicability to each case, omissions were allowed (i.e., in Israel, protest art items were cut because they did not emerge in the qualitative research, and intergroup contact items were cut because it did not occur). This set of items balances generalizability across cases with sensitivity to local nuances in youth peacebuilding.

## Phase 2: Scale development (Study 2)
**Exploratory factor analysis (EFA).** The planned analyses utilised factor analyses to explore the structure of the scale and work towards parsimony. Using Northern Ireland as the baseline case due to its relative success in peacebuilding history widely recognized in political science[61–63], we applied exploratory factor analysis (EFA) using a multi-step approach, the recommended method for developing a new scale[86]. Three principles guided scale development. First, we hypothesised a 3-factor structure reflecting the types of peacebuilding identified by the DPM[9]. Second, we aimed to balance parsimony with retaining at least 3 items for each subscale. Third, we focused on robustly capturing macrosystem peacebuilding due to a previous gap in the literature[60,102].

EFA was conducted with the psych package in R with oblimin rotation, allowing for factors to be correlated and 6 factors were identified with eigenvalues greater than 1. Parallel analysis was used in conjunction to support the most suitable factor solution. Normality for all variables was examined with skew and kurtosis (Supplementary Table S35). Maximum likelihood estimation with robust standard errors (MLR) was employed to account for non-normality, and missing data were addressed using maximum likelihood (ML) (see Supplementary Fig. S1 for a flowchart of the EFA process). This analysis was pre-registered, although we do not find the anticipated two-factor structure (see below).

First, we found that 1- and 2-factor models were not a good fit to the data[103], while the structures in the 5- and 6-factor models include a number of very small or single-item factors (Supplementary Table S9). Thus, although parallel analysis suggested a 6-factor solution for this initial pool of 22-items, toward parsimony and examining factor loadings across models, we considered the 3- and 4-factor models (Supplementary Tables S10, S11) and removed items based on both lower factor loadings for each subscale (<0.3) and theoretical validity (i.e., if they fit under the same factor conceptually based on previous literature and the DPM) and contribution, resulting in an 18-item scale.

Second, we repeated the steps for EFA, this time identifying 4 factors with eigenvalues greater than 1. The 1-, 2-factor, and 3-factor models were not a good fit to the data (CFI < 0.900, RSMEA > 0.60; Supplementary Table S12). Furthermore, although the 3-factor solution roughly followed the three types of peacebuilding, results also revealed low loadings of the voting/politics items (Supplementary Table S13). The 4-factor model was an adequate fit (CFI = 0.907, RMSEA = 0.064) and was also the solution suggested by parallel analysis (Supplementary Table S14). Examining the factor structure revealed that the items initially theorised to be macrosystem level peacebuilding appeared to split into two factors: one about voting and political participation, and the other about broader, general peacebuilding beliefs.

Third, toward parsimony and to improve model fit, five more items were reduced through repeating the previous process, resulting in a 13-item scale. EFA on this 13-item scale suggested a 4-factor model as a good fit to the data (CFI = 0.966, RMSEA = 0.041; Supplementary Tables S15–17), supported by parallel analysis. Thus, although contrary to our originally theorized 3 levels, this model was chosen as the final YPBS with four subscales across three types of peacebuilding: Macrosystem (Beliefs; 5 items), Macrosystem (Voting/Politics; 3 items), Mesosystem (3 items), and Microsystem (2 items); see Fig. 2. Similarly, although the original aim was to retain 3 items under each type, the 2-item structure under microsystem significantly improved model fit, and thus was selected as the final structure for the YPBS (Table 1). All latent factors were positively and significantly correlated with each other (see Supplementary Table S7 for a summary of the rationales for each item reduction).

## Phase 3: Scale validation (Study 2)
**Tests of dimensionality: confirmatory factor analysis (CFA).** Tests of dimensionality examine whether the factor structure applies to a new sample[104]. For the YPBS to be a global scale, the 4-factor solution identified in Northern Ireland (Fig. 2) was tested for Colombia, Israel, and Switzerland using confirmatory factor analysis (CFA) with the lavaan package in R. This analysis was not pre-registered, but we include it to provide additional support for the factor structure identified in the previous EFA. The measurement models tested in Colombia and Switzerland were identical to Northern Ireland; however, two items had been omitted in Israel, so were not modelled. No other changes were made. Model fit indices indicated acceptable to good fit across Colombia ($\chi^2(59) = 351.11$, $p < 0.001$, RMSEA = 0.080, 90% CI [0.072, 0.088], CFI = 0.91, TLI = 0.88, SRMR = 0.063; Fig. 3), Switzerland ($\chi^2(59) = 124.03$, $p < 0.001$, RMSEA = 0.049, 90% CI [0.036, 0.061], CFI = 0.96, TLI = 0.95, SRMR = 0.038.; Fig. 4), and Israel ($\chi^2(38) = 257.99$, $p < 0.001$, RMSEA = 0.088, 90% CI [0.078, 0.098], CFI = 0.90, TLI = 0.85, SRMR = 0.062; Fig. 5). Across all models, all latent factors were positively and significantly correlated with each other (Fig. 6).

**Tests of dimensionality: measurement invariance.** The planned analyses utilised a stepwise approach to assess configural, metric, and scalar invariance[105,106] using the lavaan package and anova() command in R. Configural invariance was tested by fitting an unconstrained model to evaluate whether the factor structure was consistent across cases or subgroups. Metric invariance was assessed by constraining factor loadings to equality, determining if the construct was perceived similarly. Scalar invariance was tested by constraining both factor loadings and item intercepts to equality, ensuring that any differences reflected the latent construct rather than measurement artifacts. If full invariance was not achieved, partial

**Fig. 2 | Final model with factor loadings for Northern Ireland created with 13-item YPBS with four subscales across three types.** Standardised estimates are reported.

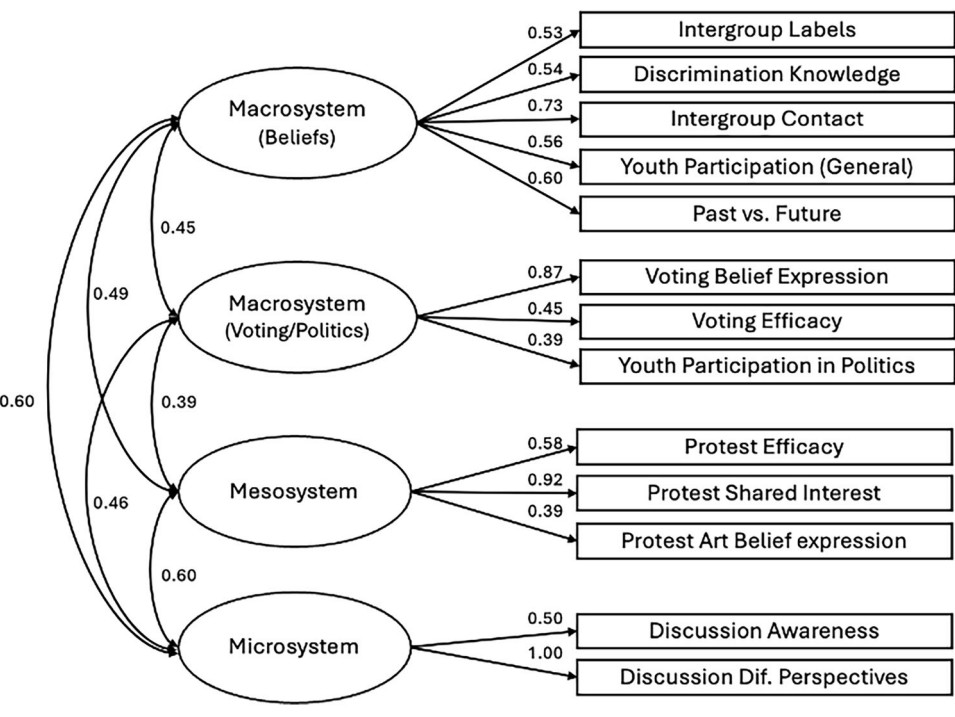

**Fig. 3 | CFA model with factor loadings for the YPBS in Colombia.** Standardised estimates are reported.

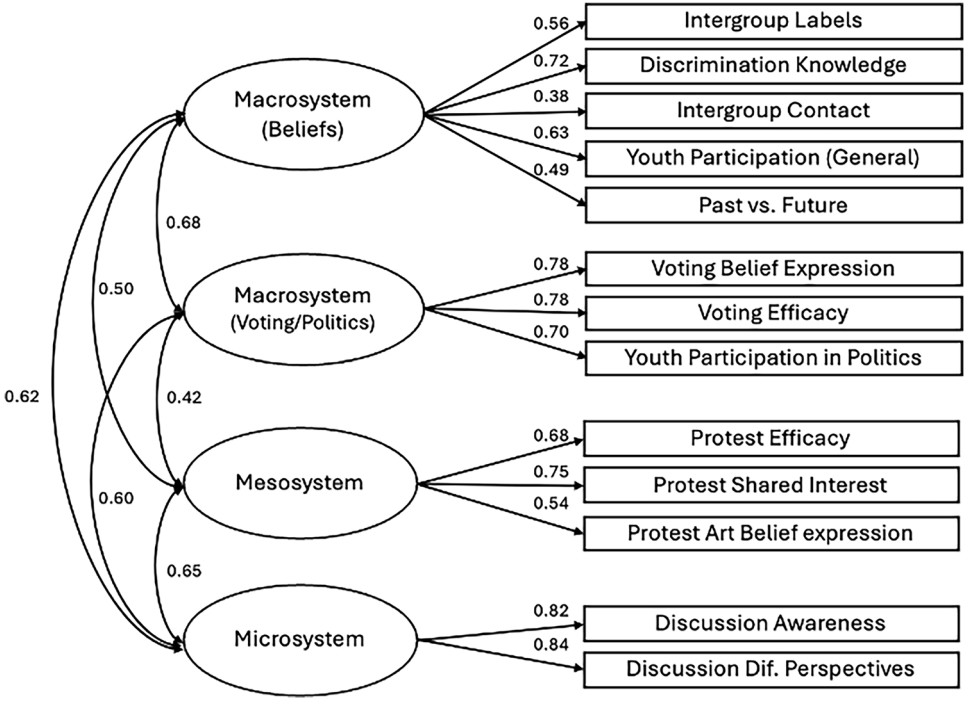

invariance was explored by systematically allowing given parameters to vary and identifying if the model fit worse. For a detailed documentation of the parameters freed to achieve partial invariance, see Supplementary Table S24. This analysis was pre-registered.

Age was categorized into two groups, adolescents and young adults, based on the typical age range associated with secondary and tertiary education within each cultural context. Gender invariance was tested with participants who self-identified as either male or female. In the case of Northern Ireland, intergroup identity was coded according to participants'

self-identification as either Protestant or Catholic. Participants who did not identify as one of the specified groups above were excluded from analysis.

First, we found support for configural invariance across all comparisons, indicating equivalent factor structure regardless of group membership within and across cases (Supplementary Table S24).

Second, across cases, measurement invariance was tested by doing pairwise comparisons to Northern Ireland (i.e., the baseline case). Partial metric invariance was achieved after freeing three parameters for Colombia (two mesosystem items and on macrosystem belief item) and

**Fig. 4 | CFA model with factor loadings for the YPBS in Switzerland.** Standardised estimates are reported.

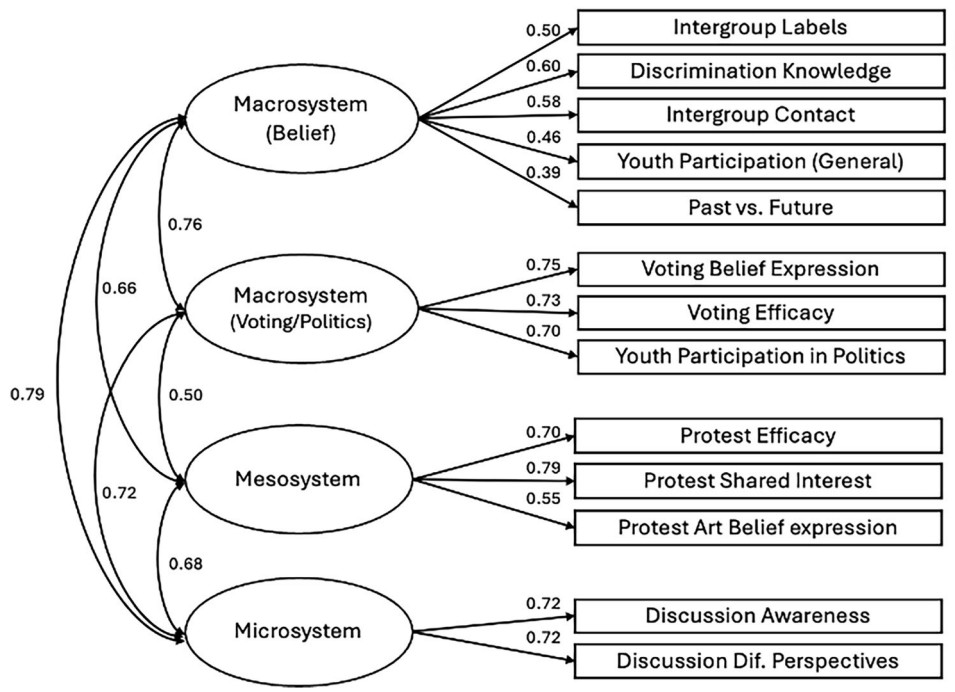

**Fig. 5 | CFA model with factor loadings for the YPBS in Israel.** Standardised estimates are reported.

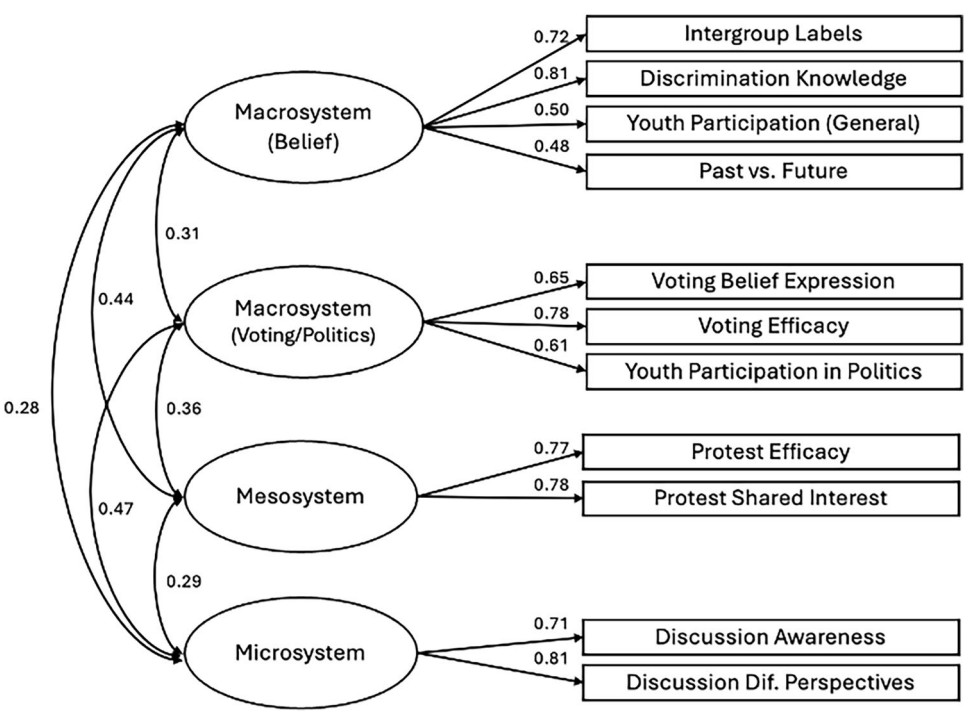

full metric invariance was achieved for Switzerland (Supplementary Table S25). Partial scalar invariance was also achieved with these two comparisons after freeing a larger subset of items. Partial scalar invariance was not achievable when comparing Israel to Northern Ireland (Supplementary Table S25). There was no consistent pattern of a specific item, or level of item, that appeared to be problematic for measurement invariance across different cases.

Third, across demographic groups, in Northern Ireland, full metric and scalar invariance were achieved for intergroup identity and gender, and partial scalar invariance was achieved for age after freeing two parameters (Supplementary Table S26). In Colombia, full metric invariance was achieved for both age and gender. Both achieved partial scalar invariance after freeing an intercept each (Supplementary Tables S27). In Israel, full metric invariance and partial scalar invariance was achieved for gender after freeing three intercepts (no adolescent sample; Supplementary Table S28). In Switzerland, full metric and scalar invariance was achieved for age. For gender, partial scalar invariance was achieved after freeing three parameters (Supplementary Tables S29). Similar to the across case comparisons, there was no consistent pattern in the parameters freed to achieve partial invariance.

**Tests of reliability: Cronbach's alpha and McDonald's omega.** We report both Cronbach's alpha and McDonald's omega for testing internal reliability across all cases, but note that McDonald's omega is generally considered the more robust test[107]: Macrosystem Beliefs ($\alpha = 0.73$, $\omega = 0.78$), Macrosystem Voting/Politics ($\alpha = 0.75$, $\omega = 0.76$), Mesosystem ($\alpha = .69$, $\omega = 0.72$), and Microsystem ($\alpha = 0.76$; $\omega$ not suitable for two items) all demonstrated respectable to acceptable internal reliability[108] (see Supplementary Table S18 for reliability statistics by case). This analysis was not pre-registered, but we included it to provide further support for scale validation.

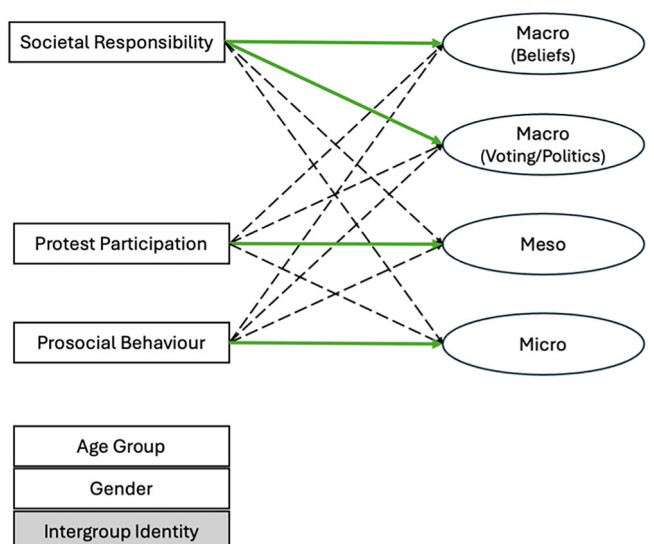

**Fig. 6 | Structural equation model tested for theorised associations across macrosystem, mesosystem, and microsystem peacebuilding.** Error variances and covariances are omitted for readability, as are paths from demographic variables (age group, gender, and intergroup for Northern Ireland; noted in grey to show not applicable across all cases) included as control variables. Paths theorized to be significant (i.e., convergent validity) are noted in green. Paths theorized to be non-significant (i.e., discriminant validity) are represented by dotted lines. All YPBS subscales are modeled as latent variables; the remaining measures are manifest variables. All tests were two-tailed.

**Tests of validity: structural equation modelling (SEM).** Using the 4-factor structure across the three types of peacebuilding, allowing subscales to covary, we then assessed validity by examining associations with other theoretically relevant measures across three levels: societal responsibility (macrosystem), protest participation (mesosystem) and general prosocial behaviour (microsystem) (see Measures section above for details). We deviated from the pre-registered plan by testing a reduced set of measures to explore convergent and discriminant validity for the YPBS for the 4-factor structure instead of the theorised 2-factor structure, which was not supported by the data.

We hypothesised that these measures would be related within level (i.e., significant paths to YPBS subscales on the same level; convergent validity), yet also distinct across levels (i.e., non-significant paths to YPBS subscales on different levels; discriminant validity)[109]. For example, convergent validity would be supported if prosocial behaviour was associated with microsystem level peacebuilding; while discriminant validity would be supported if prosocial behaviour was *not* associated with mesosystem or macrosystem level peacebuilding. Age group and gender, and intergroup in Northern Ireland, were also included as control variables.

First, we found that model fit was good to adequate across cases: Northern Ireland, $\chi^2(113) = 217.086$, $p < .001$, RMSEA = 0.042 (90% CI [0.031, 0.052]), CFI = 0.952, TLI = 0.934, SRMR = 0.037; Colombia, $\chi^2(104) = 458.107$, $p < 0.001$, RMSEA = 0.065 (90% CI [0.058, 0.072]), CFI = 0.907, TLI = 0.872, SRMR = 0.053; Israel, $\chi^2(66) = 352.345$, $p < 0.001$, RMSEA = 0.074 (90% CI [0.066, 0.083]), CFI = 0.898, TLI = 0.847, SRMR = 0.051; Switzerland, $\chi^2(104) = 206.947$, $p < 0.001$, RMSEA = 0.040 (90% CI [0.029, 0.051]), CFI = 0.957, TLI = 0.941, SRMR = 0.035. Additional analyses were conducted with socioeconomic status and political ideology as control variables but did not improve model fit (Supplementary Table S23).

Second, we found strong support for convergent validity (i.e., significant associations with the corresponding DPM level): across all cases, each measure (i.e., societal responsibility, protest participation, and prosocial behaviour) was significantly associated with the theorised type of peacebuilding (Table 2).

Third, discriminant validity (i.e., non-significant association with non-corresponding DPM levels) was less clear. Societal responsibility (macrosystem) was associated with peacebuilding across all types and cases. Protest participation (mesosystem) was associated with peacebuilding across all types in Northern Ireland and Switzerland; there was some discriminant

**Table 2 | Summary table of structural paths estimated in the SEM model**

| Case | Type | Microsystem | Mesosystem | Macrosystem (Voting/ Politics) | Macrosystem (Beliefs) |
|---|---|---|---|---|---|
| Northern Ireland | Societal Responsibility (Macro) | 0.39*** [0.21, 0.46] | 0.34*** [0.20, 0.47] | **0.45*** [0.29, 0.54]** | **0.34*** [0.15, 0.34]** |
| | Protest Participation (Meso) | −0.07 [0.03, −0.15] | **0.13** [0.06, 0.37]** | 0.12* [0.03, 0.32] | −0.03 [−0.15, 0.09] |
| | Prosocial Behaviour (Micro) | **0.24*** [0.12, 0.37]** | 0.12* [0.01, 0.28] | 0.18** [0.05, 0.34] | 0.36*** [0.20, 0.40] |
| Colombia | Societal Responsibility (Macro) | 0.38*** [0.29, 0.48] | 0.26*** [0.15, 0.35] | **0.35*** [0.27, 0.47]** | **0.48*** [0.26, 0.41]** |
| | Protest Participation (Meso) | 0.10** [0.08, 0.39] | **0.21*** [0.27, 0.59]** | 0.00 [−0.17, 0.18] | 0.04 [−0.06, 0.18] |
| | Prosocial Behaviour (Micro) | **0.24*** [0.20, 0.48]** | 0.24*** [0.16, 0.46] | 0.21*** [0.15, 0.45] | 0.32*** [0.20, 0.41] |
| Israel | Societal Responsibility (Macro) | 0.37*** [0.21, 0.37] | 0.35*** [0.33, 0.55] | **0.45*** [0.26, 0.40]** | **0.44*** [0.36, 0.55]** |
| | Protest Participation (Meso) | −0.02 [−0.14, 0.08] | **0.31*** [0.54, 0.89]** | −0.04 [−0.17, 0.05] | 0.12* [0.07, 0.39] |
| | Prosocial Behaviour (Micro) | **0.22*** [0.15, 0.38]** | 0.01 [−0.14, 0.17] | 0.22*** [0.16, 0.37] | 0.00 [−0.13, 0.14] |
| Switzerland | Societal Responsibility (Macro) | 0.30*** [0.15, 0.40] | 0.21** [0.11, 0.40] | **0.35*** [0.26, 0.52]** | **0.28*** [0.11, 0.32]** |
| | Protest Participation (Meso) | 0.12* [0.03, 0.29] | **0.34*** [0.42, 0.77]** | 0.10* [0.02, 0.30] | 0.14** [0.06, 0.25] |
| | Prosocial Behaviour (Micro) | **0.24*** [0.12, 0.37]** | 0.16** [0.08, 0.36] | 0.19** [0.10, 0.38] | 0.26*** [0.12, 0.33] |

Standardised regression estimates and unstandardised 95% confidence intervals by case. Bolded numbers depict significant paths that support convergent validity; remaining non-significant paths represent support for discriminant validity. All tests were two-tailed.
***$p < 0.001$; **$p < 0.01$, *$p < 0.05$.

**Fig. 7 | YPBS subscales by case (*N* = 2771).** Significant differences are indicated by brackets and grouped by the across-case comparison group. Whiskers represent SD. See Supplementary Table S30 for full comparison table.

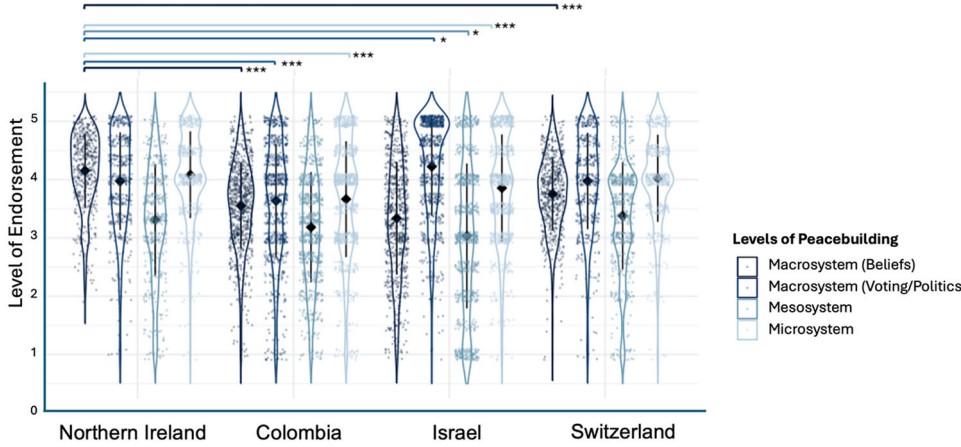

**Fig. 8 | Subgroup comparisons of YPBS subscales in Northern Ireland (*n* = 514).** Significant differences are indicated by brackets. See Supplementary Table S31 for full comparison table.

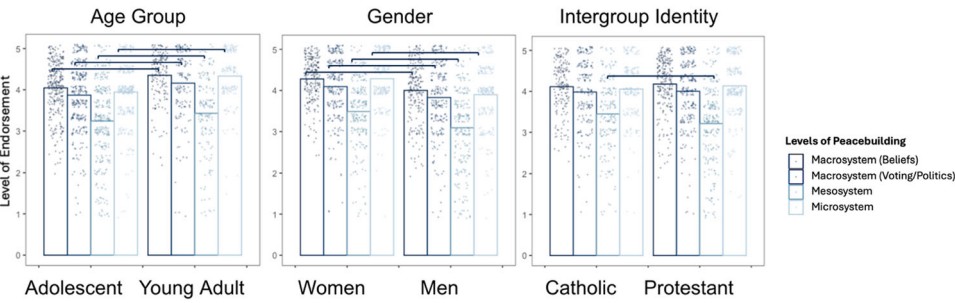

validity in Colombia (significant association for only microsystem and mesosystem peacebuilding) and Israel (significant association for only mesosystem and macrosystem (beliefs) peacebuilding). Finally, prosocial behaviour (microsystem) was significantly associated with peacebuilding across all types in Northern Ireland, Colombia, and Switzerland; in Israel, it was not significantly associated with mesosystem or macrosystem (beliefs) peacebuilding (see Supplementary Tables S19–22 for all factor loadings and structural relations).

To conclude, results supported convergent, but not discriminant, validity of the YPBS across the microsystem (i.e., relational change), mesosystem (i.e., structural change) and macrosystem (i.e., cultural change).

## Phase 4: Exploring across- and within-case comparisons with latent means (Study 2)

Because the data contained both multiple cases and multiple subgroups within each case, we explored any group differences by comparing latent means across the subscales. This was explored first across cases (Northern Ireland, Colombia, Israel, Switzerland), and second for subgroups within each case (age group and gender, with intergroup identity for Northern Ireland). These analyses were not pre-registered.

Mean comparisons were investigated by setting the reference comparison group to 0; Standardised estimates were extracted from the intercepts section of the model output. Although full scalar invariance was not established across all comparisons, we report latent mean differences for transparency and exploratory purposes. These results should be interpreted with caution, but they offer a more statistically robust and transparent alternative to traditional comparisons of observed (i.e., measured) means.

## Cross-case analyses: Northern Ireland, Colombia, Israel, and Switzerland.

Compared to Northern Ireland, Colombia showed significantly less endorsement of macrosystem (beliefs, *b* = −0.69 [−1.05,

−0.65], *SE* = 0.08, *p* < 0.001; and voting/politics, *b* = −0.58 [−0.91, −0.57], *SE* = 0.07, *p* < 0.001) and microsystem (*b* = −0.47 [−0.81, −0.49], *SE* = 0.07, *p* < 0.001) peacebuilding; Israel showed significantly less endorsement of microsystem (*b* = −0.27 [−0.49, −0.19], *SE* = 0.06, *p* < 0.001) and mesosystem (*b* = −0.15 [−0.30, −0.02], *SE* = 0.08, *p* = 0.025) peacebuilding, but significantly higher endorsement of macrosystem (voting/politics, *b* = 0.37 [0.18, 0.55], *SE* = 0.08, *p* = 0.001) peacebuilding; and Switzerland showed significantly less endorsement of macrosystem (beliefs, *b* = −0.87 [−0.96, −0.61], *SE* = 0.09, *p* < 0.001) peacebuilding (Fig. 7).

## Northern Ireland: within-case subgroup analysis.

Compared to adolescents, young adults showed significantly more endorsement across all types of peacebuilding (Macrosystem beliefs, *b* = 0.64 [0.37, 0.80], *SE* = 0.11, *p* < 0.001; Macrosystem voting/politics, *b* = 0.31 [0.08, 0.61], *SE* = 0.14, *p* < .001; Mesosystem, *b* = 0.40 [0.17, 0.61], *SE* = 0.11, *p* < 0.001; Microsystem, *b* = 0.69 [0.39, 0.80], *SE* = 0.11, *p* < 0.001). Compared to women, men showed significantly less endorsement of peacebuilding across all types (Macrosystem beliefs, *b* = −0.48 [−0.96, −0.40], *SE* = 0.14, *p* < 0.001; Macrosystem voting/politics, *b* = −0.39 [−0.66, −0.18], *SE* = 0.12, *p* = 0.001; Mesosystem, *b* = −0.55 [−0.92, −0.43], *SE* = 0.13, *p* < 0.001; Microsystem, *b* = −0.58 [−0.95, −0.45], *SE* = 0.13, *p* < 0.001). Compared to Catholic participants, Protestants showed significantly less endorsement of mesosystem peacebuilding (*b* = 0.29 [0.05, 0.47], *SE* = 0.11, *p* = 0.015), but no other comparisons were significantly different (Fig. 8).

## Colombia: within-case subgroup analysis.

Compared to adolescents, young adults showed significantly more endorsement of macrosystem voting/politics (*b* = 0.25 [0.10, 0.45], *SE* = 0.09, *p* = 0.002), mesosystem (*b* = 0.24 [0.10, 0.47], *SE* = 0.09, *p* = 0.003), and microsystem peacebuilding (*b* = 0.17 [0.02, 0.33], *SE* = 0.08, *p* = 0.032), but not macrosystem

**Fig. 9 | Subgroup comparisons of YPBS subscales in Colombia (***n* = 806**).** Significant differences are indicated by brackets. See Supplementary Table S32 for full comparison table.

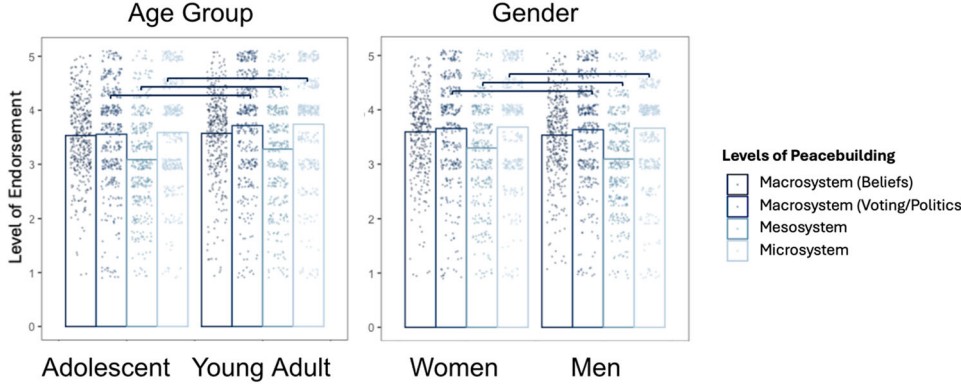

**Fig. 10 | Subgroup comparisons of YPBS subscales in Israel (***n* = 833**).** Significant differences are indicated by brackets. See Supplementary Table S33 for full comparison table.

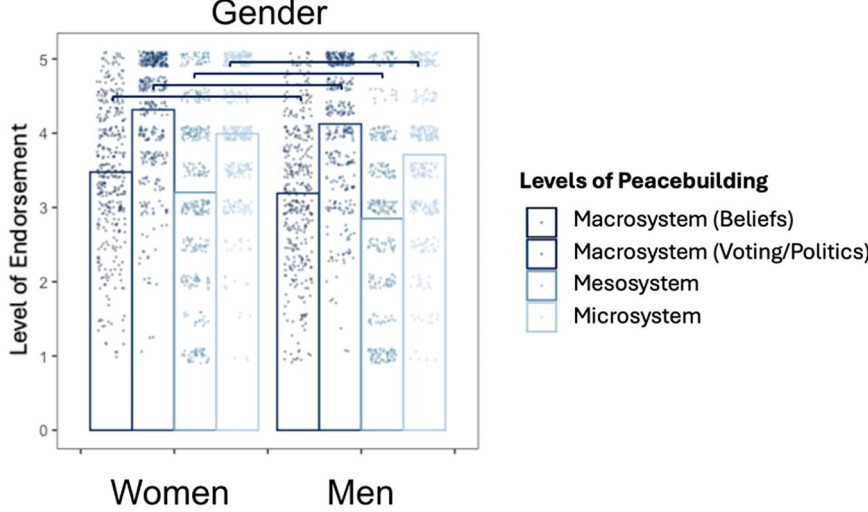

beliefs ($b = 0.07$ [$-0.09$, 0.26], $SE = 0.09$, $p = 0.345$). Compared to women, men showed significantly more endorsement of macrosystem voting/politics ($b = 0.16$ [0.01, 0.34], $SE = 0.08$, $p = 0.042$), mesosystem ($b = 0.18$, [0.01, 0.42] $SE = 0.10$, $p = 0.038$), and microsystem peacebuilding ($b = 0.16$ [0.01, 0.33], $SE = 0.08$, $p = 0.038$), but the relation to macrosystem beliefs was non-significant ($b = 0.08$ [$-0.08$, 0.27], $SE = 0.09$, $p = 0.303$) (Fig. 9).

**Israel: within-case subgroup analysis.** Compared to female participants, male participants showed significantly less endorsement across all types of peacebuilding (Macrosystem beliefs, $b = -0.19$ [$-0.34$, $-0.02$], $SE = 0.08$, $p = 0.030$; Macrosystem voting/politics, $b = -0.34$ [$-0.58$, $-0.20$], $SE = 0.10$, $p < 0.001$; Mesosystem, $b = -0.33$ [$-0.51$, $-0.17$], $SE = 0.09$, $p < 0.001$; Microsystem, $b = -0.36$ [$-0.59$, $-0.23$], $SE = 0.09$, $p < 0.001$) (Fig. 10).

**Switzerland: within-case subgroup analysis.** Compared to adolescents, young adults showed significantly more endorsement of macrosystem (beliefs, $b = 0.63$ [0.40, 0.86], $SE = 0.12$, $p < 0.001$, and voting/politics, $b = 0.78$ [0.54, 1.01], $SE = 0.12$, $p < 0.001$) and microsystem ($b = 0.32$ [0.10, 0.55], $SE = 0.11$, $p < 0.001$) peacebuilding, but were not significantly different for mesosystem peacebuilding ($b = -0.05$ [$-0.27$, 0.16], $SE = 0.11$, $p = 0.600$). Compared to women, men showed significantly less endorsement across all types of peacebuilding (Macrosystem beliefs, $b = -0.48$ [$-0.93$, $-0.36$], $SE = 0.14$, $p < 0.001$; Macrosystem voting/politics, $b = -0.21$ [$-0.55$, $-0.01$], $SE = 0.14$, $p = 0.044$; Mesosystem, $b = -0.40$ [$-0.68$, $-0.13$], $SE = 0.14$, $p < 0.001$; Microsystem, $b = -0.29$ [$-0.69$, $-0.12$], $SE = 0.14$, $p = 0.005$) (Fig. 11).

## Discussion
Addressing the call to understand how joint social action can achieve societal progress, we explored youth peacebuilding across four cases with distinct conflict histories[34]. Rooted in youth voices, we developed and validated a global Youth Peacebuilding Beliefs Scale (YPBS) across two age groups (i.e., adolescence and young adulthood) providing new insights into macrosystem peacebuilding, identifying two subscales (i.e., beliefs and voting/politics). The results have practical implications. First, used as a diagnostic tool, the YPBS can identify where youth believe social change can happen. Policymakers may adopt a strengths-based approach—one that builds on existing capacities and motivations—by directing resources toward the forms of peacebuilding in which youth already perceive opportunities for constructive engagement. Second, the YPBS is a promising tool for future large-scale international research aiming to document and understand youth peacebuilding globally, extending beyond the four cases represented.

Following best practices for scale validation[83], the lack of discriminant validity stands outs. For example, microsystem peacebuilding was significantly associated with interpersonal prosocial behaviour (theorised as microsystem), but also with protest participation (mesosystem) and societal responsibility (macrosystem). We contemplate three possibilities that may explain this finding. First, the established scales could not distinguish between the different levels of peacebuilding; other scales could be tested in future replications. Second, the types of peacebuilding are not independent. As subscales, which can be correlated, the YPBS factor structure reflects this pattern, which is also consistent in the larger prosocial behaviour literature[110]. Third, the risk or cost associated with peacebuilding in conflict settings may influence responses[111,112]. For example, societal responsibility

**Fig. 11 | Subgroup comparisons of YPBS subscales in Switzerland ($n = 618$).** Significant differences are indicated by brackets. See Supplementary Table S34 for full comparison table.

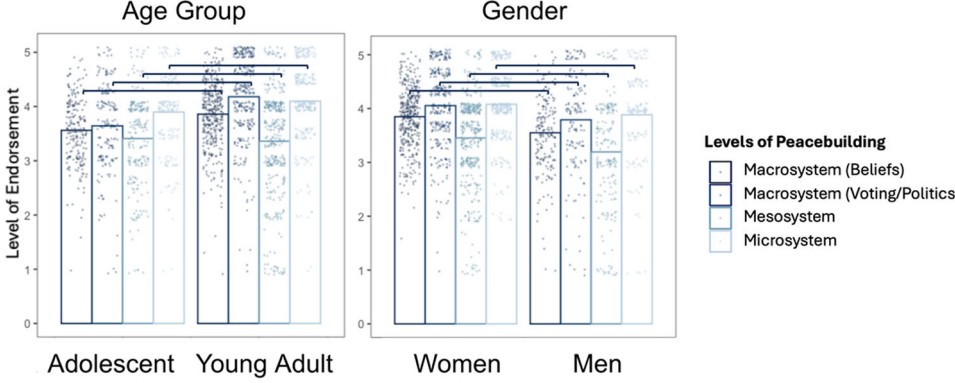

(macrosystem) was significantly associated with youth peacebuilding across all three types. Endorsing such general statements is relatively low-risk and low-cost[113], compared to helping someone (microsystem) or being involved in collective action (mesosystem). Yet, broad endorsement for peace may be a prerequisite or antecedent for higher-cost peacebuilding behaviours[53]. Sequencing studies that incorporate different timepoints, perhaps using retrospective accounting[114], might be able to tease out necessary and sufficient conditions for helping an outgroup rival or joining protests and demonstrations as an ally for the outgroup[111].

Relatedly, our strategic case selection reinforces studying the peace continuum. We found endorsement for opportunities for peacebuilding across all four cases, ranging from ongoing violent conflict (Israel), to one generation (Northern Ireland) and half a generation (Colombia) since a peace agreement, to stable democracies with no recent history of violent conflict (Switzerland). The basic YPBS structure (i.e., the items corresponding to each type of peacebuilding) applied cross-culturally; comparisons of latent means should be interpreted with some caution (e.g. which parameters freed). In Northern Ireland—one generation past peace with two decades of peacebuilding efforts— youth generally scored higher on the YPBS compared to both Colombia and Israel with two exceptions: (1) Northern Irish participants were not different from Colombian participants on the mesosystem measures and Israeli participants on macrosystem (beliefs) measures, (2) Israeli participants scored higher on macrosystem (voting/politics). At the time of data collection, endorsing voting may be especially low due to the year-long hiatus of the Northern Irish government as well as youth increasingly de-identifying with the binary identity (i.e., Protestant and Catholic) that the two-party system is based on ref. 115. Yet, one generation past peace, youth in Northern Ireland are already quite similar to those in Switzerland when considering the most effective types of peacebuilding.

We note that age and gender results were generally consistent across all cases. Young adults endorsed more peacebuilding beliefs compared to adolescents—which unsurprising given the increased autonomy, exposure, and opportunities for peacebuilding with age[116]. Women endorsed more peacebuilding beliefs compared to men (except for in Colombia), which is interesting given higher overall prosociality in women[117], but lower political participation[118,119].

## Limitations

Other limitations to the YPBS exist. For example, the reliability score of the mesosystem subscale was slightly lower compared to the others ($\alpha = 0.69$, which is below conventional standards, although the McDonald's omega was acceptable at $\omega = 0.72$). Measurement invariance testing also showed that the YPBS did not have perfect invariance across all cases and subgroups; specifically, the lack of consistent scalar invariance made it challenging to draw precise comparisons across cases. However, it is notable that across highly distinct sociocultural contexts and conflict histories, the YPBS subscales consistently mapped onto the levels of the Developmental Peacebuilding Model (DPM) and, in the broader picture, showed more

similarities than differences for its dimensionality, reliability, and validity. Thus, we recommend using the YPBS as a strong foundation for future work, with the understanding that cross-cultural applications should always incorporate validity checks—such as confirmatory factor analyses, invariance testing, and potential item adjustments. Despite these limitations, to our knowledge, the YPBS is the first to robustly assess different levels of youth peacebuilding cross-culturally, specifically with macrosystem subscales that address youth's contributions to broader societal and cultural change across the peace continuum.

## Future directions

Considering future applications of the YPBS, we identify four key next steps in this research agenda. First, adolescence represents a key point in development highlighted by increasing autonomy[120,121] and growing awareness of structural injustice[122,123]. These developmental shifts result in greater opportunities and motivations to engage in both structural change and cultural change[31], allowing us to investigate the early onset of peacebuilding across the social ecology. Future research might explore if the YPBS could be applied to adults. Second, focus groups highlight generational differences around macrosystem norms, such as how youth perceive the conflict compared to older generations. The YPBS could be used to track generational patterns in peacebuilding, complementing previous work on collective action[124]. Third, despite strategic case selection, applying the YPBS in other settings along the peace continuum and robust sampling from conflict rival or salient majority/minority groups would deepen understanding about how key power dynamics shape youth peacebuilding. Finally, although the YPBS is currently a *beliefs* scale, future research should explore measures that can capture peacebuilding *behaviour* across different levels, as well as other aspects of peacebuilding (e.g., institutional versus non-institutional participation).

## Conclusion

In conclusion, we developed and validated a global Youth Peacebuilding Beliefs Scale for adolescents and young adults. Our approach addressed current gaps in the literature by focusing on youth as agents of change and the peace continuum. We further contributed to understanding by prioritising youth voices, alongside rigorous scale development, and providing empirical support for the Developmental Peacebuilding Model[9]. More broadly, we advance conceptual understanding, engaging in the conversation about prosocial action in high-risk settings[125] and provided new ways to assess macrosystem peacebuilding. Finally, our empirical research directly supports the UN's Youth Peace and Security Agenda, providing an easy-to-use tool to assess and mark progress on youth peacebuilding around the world.

## Data availability

Qualitative data for Study 1 are not available due to sensitivity of the data. The data files for Study 2 are available at this link: https://osf.io/5bn49/.

## Code availability

The analytic codes for Study 2 are available at this link: https://osf.io/5bn49/.

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

## Acknowledgement

The research conducted in this publication was funded by the Research Ireland under grant number IRCLA/2022/3482. The funders had no role in study

design, data collection and analysis, decision to publish or preparation of the manuscript. We extend our sincere thanks to the members of the Helping Kids! Lab for their valuable feedback on the study design, data analysis, and manuscript preparation. We also thank Dr. Jocelyn Dautel and Dr. Dinah Gross for assisting with data collection. We are grateful to the following research assistants for their contributions to data collection, transcription, and translation: Chris Hastings, Murray Kennedy, Mary-Jane Emmett, Joanna Harnett, Rona Yaniv, Neta Dekel, Alon Bar-peled, Anyela Daniela Rocha Galvis, Maria Alejandra Mejía Veloza, Julian Steven Mesa Vargas, Esther Hegnauer-Camille, and Luise Victoria Rebentisch. Finally, we thank all the families, schools, and youth groups who assisted with participant recruitment.

## Author contributions

Laura K. Taylor: Conceptualization, Formal Analysis, Funding Acquisition, Investigation, Methodology, Project Administration, Resources, Supervision, Writing—Review and Editing. Vivian Liu: Data Curation, Formal Analysis, Methodology, Investigation, Project Administration, Validation, Visualization, Writing— Original Draft Preparation. Bethany Corbett: Data Curation, Methodology, Resources, Writing—Review and Editing. Juliana Valentina Duarte Valderrama: Data Curation, Investigation, Methodology, Resources, Writing—Review and Editing. Léïla Eisner : Data Curation, Investigation, Methodology, Resources, Writing—Review and Editing. Jeanine Grütter: Data Curation, Investigation, Methodology, Resources, Writing—Review and Editing, Supervision. Eran Halperin: Data Curation, Methodology, Resources, Writing—Review and Editing, Supervision. Tabea Hässler: Data Curation, Investigation, Methodology, Resources, Writing—Review and Editing. Claudia Pineda-Marin: Data Curation, Methodology, Resources, Writing—Review and Editing, Supervision. Ilana Ushomirsky: Data Curation, Investigation, Methodology, Resources, Writing—Review and Editing.

## Competing interests

The authors declare no competing interests.
