## [Transparent Peer Review file · Communications Psychology]

A Global Youth Peacebuilding Beliefs Scale

Corresponding Author: Dr Vivian Liu

Version 0:

Decision Letter:

Dear Dr Liu,

Thank you for your patience during the peer-review process. Your manuscript titled "GENERATION PEACE: Developing and Validating a Global Youth Peacebuilding Beliefs Scale (YPBS)" has now been seen by 3 reviewers, whose comments are appended below. You will see that they find your work of some potential interest. However, they have raised quite substantial concerns that must be addressed. In light of these comments, we cannot accept the manuscript for publication, but would be interested in considering a revised version that fully addresses these serious concerns.

We hope you will find the Reviewers' comments useful as you decide how to proceed. Should additional work allow you to address these criticisms, we would be happy to look at a substantially revised manuscript. If you choose to take up this option, please highlight all changes in the manuscript text file, and provide a detailed point-by-point reply to the reviewers.

Editorially, we consider it critical that all methodological concerns voiced by our scale-development experts (Refs #2 and #3) are comprehensively addressed. These concerns are not only presentational and will require significant revisions.

At the same time, the presentation also needs to be improved. Please note that Communications Psychology publishes Articles in which the Methods section follows the Introduction and precedes the Results. No word limits apply to the Methods and Results sections and all critical elements should be in the main paper, not the Supplementary Information form.

I am attaching a checklist that details critical reporting requirements for the revised manuscript. Please attend to each item and ensure your manuscript is fully compliant. We are requesting that your manuscript aligns with these requirements as this facilitates the evaluation of your manuscript, reducing delays in re-review and potential future acceptance. If your revised manuscript is not aligned with these requests on major issues, such as those concerning statistics, it may be returned to you for further revisions without re-review. Additional information can be found in our style and formatting guide Communications Psychology formatting guide.

If the revision process takes significantly longer than five months, we will be happy to reconsider your paper at a later date, provided it still presents a significant contribution to the literature at that stage.

Please use the following link to submit your

- revised manuscript,
- point-by-point response to the referees' comments,
- cover letter (as a separate document),
- the Editorial Policy Checklist (see below),

- the Reporting Summary (see below), and
- the completed Editorial Request Table (attached):

Link Redacted

Thank you for the opportunity to review your work.

Best regards,

Marika Schiffer

Marika Schiffer, PhD
Chief Editor
Communications Psychology

REVIEWER REPORTS:

Reviewer #1 (Remarks to the Author):

In "GENERATION PEACE: Developing and Validating a Global Youth Peacebuilding Beliefs Scale (YPBS)," the authors present the results of validating a new scale measuring peacebuilding beliefs among young people in four country contexts with different experiences with violent conflict. Their approach is unique to the extent that it focuses on youth's peacebuilding potential, rather than looking at youth as actors of violence, and to the extent that it looks simultaneously at countries currently experiencing both violence and peace. I deeply appreciate their efforts and the scope of the work they have done. That having said, the article would benefit from revisions which focus more on the content of the scale. In what follows, I detail six areas for improvement which need to be addressed in my opinion prior to publication.

- Deeper engagement with the existing literature. The authors rightly point out that in the past, the literature was very focused on youths as vandals, rather than vanguards, and that they typically focus either on a conflict-affected society, either on a society in peace. However, their discussion of this literature is very brief. It also fails to acknowledge that more recently (though key articles are from the early 2000s already), scholars increasingly recognize youth's peacebuilding potential. More engagement with these studies would be useful, particularly looking at the contexts that are under study in their study as well (i.e., Israel, Switzerland, Northern Ireland and Colombia). Such a review could also shed more light on the existing measures. The references to the other scales in the section on discriminant validation come out of the blue. What other scales do exist (even if not focused on youth in particular)? What are their shortcomings? And, thus, why do we need this scale?
- More background to the current study: there is very little background information. What was the rationale of this study? This relates back to a discussion of other measures. Is it that no other measures exist? But also, what exactly do the authors wish to measure with this scale? Is it youth support for particular types of peacebuilding activities? Their peacebuilding beliefs? How do they relate the UN agenda that is mentioned in the intro? Second, why were precisely these four contexts chosen? In this respect, the article also glosses over recent violent youth protests in the case of Northern Ireland. Third, how were youth selected? Admittedly, some criteria are shared, but they are very broad begging the question how youth were approached and thus who ultimately participated.
- Include insights from focus group discussions: clearly, a lot of work was invested in developing the scale. Having more information on the FGDs would enrich the article significantly. This includes more information on the questions that were asked, but also, and most importantly, some of the responses of the young people who participated. In presenting some quotes, you could also reflect on emerging differences between youth.
- 3 types of peacebuilding: I appreciate the idea to distinguish between types of peacebuilding, but I struggle with the division into micro, meso and macro. Whereas the micro and meso level are clear, the macro level supposedly refers to beliefs. However, why would beliefs be on a different level than structural change? I would situate both cultural and structural change at the same level. There could be a difference between the national and the international context, but here is no mention of. Would it be more useful to follow Galtung's distinction between direct, structural and cultural violence and accordingly have direct peace in the young person's life (now micro), structural peace (now meso) and cultural peacebuilding (now macro level)? Importantly, calling beliefs macro level also seems to bypass the importance of context.
- Presentation of statistical results: Finally, I invite the author(s) to interpret their results theoretically, and to reflect upon the reasons why at times intercepts had to be released, etc. Freeing the intercepts is fine, but only if it makes sense theoretically.
- Structure: The structure of the article is peculiar, with the methodological section coming very late.

I wish the author(s) the best of luck with their revisions.

Reviewer #2 (Remarks to the Author):

"GENERATION PEACE: Developing and Validating a Global Youth Peacebuilding Beliefs Scale (YPBS)"

"To be precise, we would sincerely value your feedback given your expertise in scale development. A short report focusing only on this aspect would entirely suffice for our purposes, as we have other subject-matter experts comment on conceptual issues."

Review

I have read the whole papers, as I had to do to make any sense of the primary request to me. There is clearly a huge amount of work in this study, and the topic is original and important. Nevertheless, there are some significant issues with the article that I strongly suggest need attending to if this paper is to stand up to the scrutiny and scientific quality required in a Nature journal.

That is, corrections / revisions are required.

I will focus on the task I have been given in my report, then summarise using the given questions.

Structure: The unconventional presentation of the sections of this research is a challenge for any reader. There is a need to have the Methods section of a scale development paper, where methods are from and centre before the results and discussion, in the article after the Introduction. Presenting the main methods section at the end, of the article, and then more Methods in a supplementary file as a kind of optional extra for reporting the research, makes for a hard read. This must be addressed.

Focusing on the methods, the mixed methods, and the scale development:

1. I would expect the authors to have specifically stated that they were using a Sequential Exploratory Mixed-methods design to develop the study – and to have included a visual / figure to identify the phases of the tool develop with procedures and products. Whilst there is some semblance of this in figure 1, the details are sparse.

2. Related to the above, the information in Phase 1 is far too succinct (especially in what one reads first, in the Results section, but even after visiting the end section, and the supplementary material).

a. There is no information of use of the literature in Phase 1, how it informed the interview guide for the focus groups beyond the (assuming it did), beyond sourcing the Developmental Peacebuilding Model (DPM) as a theoretical framework,

b. The practice associated with recruitment for the focus groups is inadequate.

c. Details on how the Focus Groups were conducted is vague. It is clear that there was a not consistent and this was not only across cases.

d. Information on the analysis is inadequate (see below) but particularly regarding how codes from the qualitative analysis were translated into the initial items for the Qualitative phase.

I suggest that some of the supplementary information referred to in the text is actually his is basic information, and it would be helpful to be included as a part of the main article.

3. As the study used the Developmental Peacebuilding Model (DPM) as a theoretical framework for the scale development, I suggest that a Directed Content Analysis is more appropriate than Braun & Clark's Thematic Analysis used on the collected qualitative data. One has to assume the themes found would emerge from what was asked. The interview guide was based on the DPM – as I learned from the Supplementary Material, and line 363 it is noted that this lens was used, so why Thematic Analysis?

4. Related to the above, the use of Exploratory Factor Analysis (EFA) mentioned in the Introduction (last paragraph) was not the driver for the initial item reduction, but the use of experts. An Experts Panel is used in a study basing scale development on a theoretical framework to help establish face and content validity, to ensure the items kept in the developing tool adequately reflect the constructs of interest (here known from the DPM).

The procedure for the expert consultation is completely missing. The approach is correct, as one would expect an Experts Panel to be involved in the initial consideration of items one would expect to see in such a scale, and what has emerged from the codes to be developed into items, and then refinement in the number of items. In no part of the article is there information regarding the activity of the experts and what did they do to reduce the initial 52 items in the developing scale to 22 items.

Assuming this was conducted – the article should provide information on the above, and provide the CVI and CVR from the experts' ratings. It would also be usual to state how many experts were used. It would also be usual to refer the number of items to the 3 dimensions of the DPM.

5. Phase 2 includes the previously indicated EFA to explore the potential underlying structure of items in the reduced 22-item scale. Returning to the narrative in the introduction, of course, you would hypothesise a three-factor structure: the whole data collection was steered that way - as would be expected based on three-factor framework (the DPM). In practice, we are told that the EFA indicated a four-factor structure with more than half the final 13-item scale associated with the macrosystem in two scales. With this sample size, a split-half EFA and subsequent reliability of the outcome could have been performed. The outcome is unsatisfactory in terms of balance. The resultant 2-item microsystem subscale left this reader wondering if the EFA was used appropriately, especially noting the aim to retain at least three items for each subscale (see line 408, and notwithstanding lines 433-435). It seems some items may have been removed from a scale, not because of the loading, but because of the theoretical fit. This is not really in line with EFA.

I recommend a split-half procedure is performed on the data as a check-back, (after looking at the Phase 1 procedure again regarding initial item development if necessary).

Table 1 (or a revised Table 1) should be furnished with factor loadings and the Alpha co-efficient for each scale. Related to the above, the Mesosystem subscale is not reliable, despite the reference given. DeVellis & Thorpe (2021) discuss reliability and alpha in some length in the given reference. They rightly point out some limitations, and no definitive thresholds, nevertheless, their self-professed utilitarian favour given to using alpha to demonstrate reliability indicates thresholds above

.69.

6. Some explanation is required to illuminate why two items from the EFA were not applicable to one of the four cases – how did that emerge? Critically, how does this align with the aim that the YPBS is a global scale?

7. The account of the Confirmatory Factor Analysis could be clearer. There seems to be an assumption that the Northern Ireland model was suitable and sufficient to be measured against (even in line 437). Associated with the above, there are no figures given for Israel regarding the CFA: only Colombia and Switzerland, although all three are in Table S26.

8. The account of Measurement Invariance also needs some outcome results to be meaningful.

9. Tests of validity may show adequate validity, as argued. Could they be better for returning to the data, and considering the data reduction steps more rigorously. Unfortunately, the lack of information does not allow me to suggest this is necessary, but what I can see is an unjustified reduction from 52 items to 13 items, a subscale that is unreliable, and a global scale that includes two of the 13 items which are described as omitted from one of the countries / cases. These are issues that must be resolved for the YPBS to be a useful diagnostic tool.

10. Whilst the discussion argues for best practices in scale validation, that it is “rooted in a conceptual framework, the Developmental Peacebuilding Model” is challenging.

11. Overall, there are several tense issues that should be addressed.

Regarding the given questions might assist you in writing a helpful, well-justified review.

-Does the paper represent an advance in understanding which may influence thinking in the field? If you have concerns about the advance in relation to specific studies, we appreciate references to this work. It should do, however I suggest further work needs to be undertaken to ensure the research is clear and its arguments are valid.

-Does the article presents an original study, new analysis, new model, or a direct or extended replication of previous work?
Yes

-Are the data and analysis technically sound? Are they appropriate to answer the research question, e.g., are causal research questions addressed on the basis of causal, rather than correlational evidence?
No. The issue could be poor structure, and gaps in explanation, nevertheless, I am concerned about the methodology in scale construction. Specifically, the analysis of the qualitative data, and its integration to develop items, the initial consideration of the validity of those items, the Experts Panel that was used for the initial reduction of items, the use of EFA, and tests of reliability. I am also perplexed regarding what is essentially an 11-item version of the scale for one country and 13-items for another.

I recommend revisiting some of the analyses if the scale is to truly be a global scale, and one that is to be useful for the purpose of peacebuilding. This may need more items and more attention to the theoretical framework it was using.

-Does the paper provide strong evidence for its conclusions?
Not strong.

-Is the study question important to scientists for a sub-field of psychology?
Yes.

-Are there any special ethical concerns arising from the use of animals or human subjects?
No.

- Was the study preregistered and if so, did the authors follow the preregistration?
Not checked.

Rosanna Cousins
07.05.2025

Reviewer #3 (Remarks to the Author):

The submitted article “Generation Peace: Developing and validating a global youth peacebuilding beliefs scale (YPBS)” contains multiple studies across a variety of cultures, and includes more difficult to collect adolescent samples, detailing the item development and then psychometric testing (i.e., EFA, CFA, tests of validity) of the retained 13 items that create a four factor measure of peacebuilding. Overall, the paper contains a wealth of analyses, of which most of the statistics are reported in the supplementary materials, to support the item generation and confirmation. The tests of convergent and discriminant validity seem a bit mixed, which does leave some lingering questions about the validity, and the current structure of the paper made evaluating the methods and interpreting analyses cumbersome. Below are suggested minor and major revisions that may help strengthen the paper for a resubmission for potential publication.

Major Revisions:

Overall, my biggest suggestion would be to reorganize the structure of the paper so that more of the statistical analyses are directly reported in the main text. I found it odd to finish reading the discussion and then find the methods for Study 1 and 2 reported. I thought perhaps this may be a suggestion of formatting from the journal, but I read some other recent publications

and these seemed to follow the standard intro, methods, results, discussion format.

I understand there are a lot of additional analyses and greatly appreciate these reported in the lengthy appendix, but I found the current draft lacked sufficient methodological detail in the main text to help support the claims. Again, it's not that the details were totally absent – the empirical support for the claims is provided, but just not easily found. I'll detail some more specifics below, but one example would be in line 103 where the Study 1 sample of 199 participants is noted, the reader has to go both lines 330-337 and the supplementary materials to find more information about the breakdown of the sample size by country, and the sample demographics. This made following the article difficult, as I kept having to shuffle across pages to learn about the sample. I'd suggest instead incorporating the lines 330-337 into the section in line 103, and then also adding the young adult vs. adolescent sample size and percentage breakdown found in table S2 within a sentence reported around line 103. In another instance, I was left wondering what estimation method and rotation were used in the EFAs.

Thus, I think the methods detailed from lines 329-464 would be better incorporated in their respective spots described earlier in the paper.

Line 127: please note the likert scaling anchors used for the scale items here. Likewise, some of the sample demographics here would be helpful, and clarification on what "Survey company" was noted in TS5, since other survey companies like Prolific, are noted by name.

Line 132: The "Kaiser criterion" of using an eigen value greater than 1 as a metric for determining the number of retained factors has been shown to be a fairly arbitrary guideline that can lead to under or overfactoring based on features of the data (see Preacher & MacCallum, 2003 for a detailed discussion). I'd recommend instead that a parallel analysis be conducted so that the observed eigenvalues are compared to those randomly generated.

Line 135: a bit more is noted in the supplementary materials, but it should be described more here in the main text a how the scale was reduced from 22 to 18 to a final 13 items.

Were any of these scales translated to Spanish or other languages based on the sample the scale was administered to? Please describe if so, or if the items were all administered in English in the Phase 2 section.

Tests of dimensionality: please clarify that it seems the tests of measurement invariance were conducted pair-wise, with Northern Ireland as the reference group and Colombia, Israel, and Switzerland as the focal groups. It does not appear that the invariance testing was conducted across all 4 groups at once (which is a possibility in invariance testing).

Table 1: Items 1 and 2 for the macrosystem subscale involve voting, but the third item (we need more young people in politics) really appears to not be voting at all, and instead is more of a measure of political involvement. Would a better name for the factor be "youth political involvement" or "political activity" ?

Throughout both the EFA and CFA results, please discuss the correlations between the 4 factors.

Lines 156: Please describe how the different groups of age, and gender and "intergroup identity" were specifically defined for these invariance tests.

Throughout the measurement invariance section the details on establishing the partial invariance seem rather light. Which items require freeing the intercepts or factor loadings? Were any items particularly problematic for a subgroup?

Tests of reliability: Are these measures of reliability calculated across all samples and demographic groups examined? Coefficient alpha assumes tau-equivalence across the items. A additional reporting of Omega may be useful.

Tests of validity: For these tests of validity more information should be provided describing these related constructs (e.g., social responsibility), how these were measured (some is noted later at the end of the paper), and cite research for why these would be theoretically related (convergent validity) or uncorrelated (discriminant validity). What were the specific theories used for examining these tests of validity and what hypotheses were formed? In particular, what theoretical support is there to have measures of prosocial behavior predicting peacebuilding beliefs. Other theories may suggest instead that peacebuilding beliefs should predict prosocial behavior as the outcome.

Lines 194-202: I was a bit confused here in the tests for discriminant validity, but this may be because I was unsure which constructs/measures were hypothesized to not correlate with the peacebuilding subscales. For example, in line 198 it's discussed which variables significantly predicted the peacebuilding constructs in the Colombian sample, but discriminant validity is actually established when the scale is not correlated with other distinct constructs. I was left feeling unconvinced that the 4 subscales are distinct from other measures. Likewise in the Discussion around lines 273-278 further clarification would be helpful.

Figure 1: Please note in the text if these predictors are all measured at the manifest level instead of as latent constructs. Are age, gender, and intergroup covariates? If so please note this in the note for the figure.

Table 2: The note describes that bolded numbers depicted support for convergent validity, but is there any way to note if any of the noncorrelations support evidence for discriminant validity?

Cross-case analyses: These tests of mean differences appear exploratory. If so, please note as such. Otherwise, please provide past research or theory for the differences one may hypothesize (e.g., would it be hypothesized that Colombia would show lower macrosystem compared to Northern Ireland?). Is there any past research for gender differences or age differences?

Cross-case analyses: Throughout this entire section, please note more specifically what statistical test is used and report more of the statistics than just the p-value. For example means and standard deviations for the groups would be helpful. I was left confused whether the p-values were from t-tests, f-tests, or latent mean equivalency tests based on the change in chi-square from the constraint present or not. In line 294 latent means were mentioned, but I don't think the test was mentioned when reporting the results.

Minor Revisions:

Line 6: please note what the acronym SDG stands for

Line 31: when 1/3 of the population is mentioned, is it mean youth are 1/3? If so please clarify.

Line 34: are there other examples of nonviolent civic engagement that could be added to this e.g.?

Lines 40-46: Are there any past measures of peacebuilding? Presumably these were developed with adult samples if they exist, but please detail some past measures so readers are aware how this measure differs.

Line 55: please provide a clearer or more detailed example of what is mean by "civic engagement for specific groups"
Line 55-59: please note the difference between direct and structural violence. The term is also mentioned in line 76.
Line 64: this sentence on macrosystem may work better at the start of line 60.
Line 113: should this sentence end with "of peacebuilding"?
Line 133: please define theoretical fit a bit more.
Line 134: please explain a bit more what is mean by "heightened focus on retaining macrosystem". To me, I interpreted this as the researchers just really wanted ot keep the voting items no matter what.
Lines 147- 152: Please note the sample sizes for the different countries here. Please also note in like 152 it's tables S26-S30 for the results.
I may have missed it, but it seems table S31 is not mentioned anywhere.
Line 181: Do the authors mean Figure 1 instead of 2?
Line 268: please define "strengths-based approach".
Line 277: please define "distinct types". Is this peace building types?

EDITORIAL POLICIES

We ask that you ensure your manuscript complies with our editorial policies and reporting requirements.

To that end, we require revised manuscripts to be accompanied by two completed items: a reporting summary that collects information on study design and procedure, and an editorial policy checklist that verifies compliance with all required editorial policies

- <https://www.nature.com/documents/nr-reporting-summary.zip>>Nature Research Reporting Summary
- <https://www.nature.com/documents/nr-editorial-policy-checklist.pdf>>Editorial Policy Checklist

All points on the policy checklist must be addressed. Your revised manuscript can only be sent back to the referees if these checklists are completed and uploaded with the revision.

Notes: If you have submitted a Stage 1 Registered Report, Review, Primer, Comment, or Perspective you do not need to submit these forms. If you have already submitted these forms, you may disregard this request.

* TRANSPARENT PEER REVIEW: Communications Psychology uses a transparent peer review system. This means that we publish the editorial decision letters including Reviewers' comments to the authors and the author rebuttal letters online as a supplementary peer review file. However, on author request, confidential information and data can be removed from the published reviewer reports and rebuttal letters prior to publication. If your manuscript has been previously reviewed at another journal, those Reviewers' comments would not form part of the published peer review file.

** Visit Nature Research's author and referees' website at <http://www.nature.com/authors>>www.nature.com/authors for information about policies, services and author benefits**

If you experience problems in linking your ORCID, please contact the <http://platformsupport.nature.com/>>Platform Support Helpdesk.

Version 1:

Decision Letter:

Dear Dr Liu,

Your manuscript titled "GENERATION PEACE: Developing and Validating a Global Youth Peacebuilding Beliefs Scale (YPBS)" has now been seen by our reviewers, whose comments appear below. In light of their advice I am delighted to say that we are happy, in principle, to publish a suitably revised version in Communications Psychology.

We therefore invite you to revise your paper one last time to address the remaining concerns of our reviewers and a list of editorial requests. At the same time we ask that you edit your manuscript to comply with our format requirements and to maximise the accessibility and therefore the impact of your work.

EDITORIAL REQUESTS:

Please review our specific editorial comments and requests regarding your manuscript in the attached "Editorial Requests Table". Please outline your response to each request in the right hand column. The most fundamental outstanding issue are deviations between the preregistration and the final analysis in the paper. More guidance on the required revisions are included on the Editorial Requests Table. Please upload the completed table with your manuscript files as a Related Manuscript file.

SUBMISSION INFORMATION:

OPEN ACCESS:

* DATA AVAILABILITY:

Link Redacted

Best regards,

Marika

Marika Schiffer, PhD
Chief Editor
Communications Psychology

REVIEWERS' COMMENTS:

Reviewer #1 (Remarks to the Author):

I thank the authors for the careful revision of this important paper. Once again, I have read the paper with great interest. As outlined in the rebuttal letter, many suggestions of the reviewers have been taken at heart. I particularly applaud the restructuring of the paper, the inclusion of information and quotes regarding the content of the FGDs and survey, as well as the inclusion of additional literature. The revision also sheds more light on the theoretical decisions that were made, that were not entirely clear to me when reading the paper for the first time.

While I consider the revision largely satisfactory, I would still like to share the following suggestions for further improvement:

- Characteristics of respondents: whereas more information is shared about the FGD samples in each context, I would include the gender distribution for each country *in text* - just like the adolescent- young adult divide, the gender divide is crucial because also used further in the analyses. This information could also be added for the survey in text.
- At the macrosystem, a difference is made between voting/political participation, and more general peacebuilding beliefs. It could be useful to refer here to the literature on youth political participation, which distinguishes between institutional (voting, party membership, etc.) and non-institutional participation (protests, boycotts, etc.). A direct parallel cannot be made, but it is useful to reflect on the similarities and differences. Further, with respect to the general peacebuilding beliefs, I am not convinced that all these items measure what they intend to measure (intergroup contact is usually considered at the interpersonal level). This is more difficult to address, but some reflection at some point would be useful.
- The Across- and Within-Case Comparisons remain very descriptive. There is hardly any interpretation of the results. This should not necessarily be added in the analysis, but at some point in the discussion a reflection on the gender and age differences is warranted, however short. Relatedly, I miss a reflection on the type of conflict that affected the country (inter- or intra-country violence).
- Finally, there are some typos that remain. Consider for instance, missing comma at sentence 436, incomplete sentences at 509-510.

Reviewer #2 (Remarks to the Author):

I have read the rebuttal, the revised paper and the supplement. The authors are to be congratulated for grasping the significant amount of work that needed to be done to bring it up to the current publishable standard. That said, I suggest the small number of points I note below should be attended to.

1. I previously noted (point 10) that within the article there are several tense issues that should be addressed. R2.11 suggests that tense errors have been attended to. There remains, in the introduction, however, several tense errors. I.e. the present or future tense is used, not the past. For example:
Line 21 examine should be examined
Line 23 develop and validate should be developed and validated (as at line 578)
Line 27 address should be has addressed; similarly line 45 address should be addressed
Line 71 will serve should be served
There are more.
2. I appreciate that the data was collected before October 7th 2023, nevertheless, the assertion at lines 180-1 that protest art rarely occurred in Israel could be contested, even at the point of collection. As this article is likely to be published in the context of much protest art in Israel, perhaps there should be acknowledgement of this, and justification for the withdrawn question (Art 2) being included in any subsequent promulgation of the YPBS in Israel.
3. Better clarity is needed in the methodology insofar as the scale used in line 192, has yet to be developed according to line 204.
4. Table S7 needs revising. The table indicates that 14 items have been retained, yet the associated text and Table gives 13 items. The 'lost' item is Protest 1 "I feel inspired by protests". There is also a different items called Protests 1. Assuming that Protest 1 should be Protests 3, the factor loading is .65, and it is in the final version (Table S25), but along the way through the other tables, it is difficult to see why this should be and / or what has happened to Protests 1. Altogether, for this reader, the sheer volume of tables in the supplement is not illuminating but confusing. I suggest these are checked and / or slimmed down with a succinct narrative to justify the item-reduction process.
5. Other typos should be corrected at lines 326, 369, 432, 480, and 519-20.

Reviewer #3 (Remarks to the Author):

I have reviewed the authors' rebuttal letter and revised manuscript. I appreciate the authors' careful attention to my review. This revised manuscript contains a better organized structure and clearer details on analyses conducted. I'm satisfied with this revision and have no additional requests.

Dear Reviewers,

Thank you for the opportunity to revise our manuscript, , “GENERATION PEACE: Developing and Validating a Global Youth Peacebuilding Beliefs Scale (YPBS)”, to be submitted to *Communications Psychology*. We sincerely appreciate the thoughtful and constructive feedback provided, and have carefully considered all comments.

We outline below the major revisions made to the manuscript (noted in blue text in the main manuscript). We also have moved critical information from the supplement to the main paper and restructured the order of sections (noted in green text in the main manuscript). We would be happy to incorporate more supplementary content if desired, but have aimed to balance thoroughness with conciseness considering the journal’s word limit of 5000 words.

Reviewer 1 (R1)

We thank R1 for recognizing the unique contribution of studying youth’s peacebuilding potential and the scope of this work.

We made the following revisions based on their feedback:

1. Deeper engagement with the existing literature. We have expanded several points in our introduction; we have cited almost 2 dozen articles and books, including 10 from the last 5 years. We also have added over a dozen articles that specifically focus on Israel, Switzerland, Northern Ireland, and Colombia. We welcome additional suggestions we may have overlooked.
2. More background to the current study. We have more explicitly noted the unique contributions of the YPBS: tested cross-culturally in four cases across the peace continuum, across two developmental periods, and covering peacebuilding at three levels. We also have clarified the goals and content of the YPBS, connecting to UN agendas of recognising youth peacebuilding as multi-level. Case selection has been strengthened, from a theoretical perspective and with additional citations (see R1.1), and we have also expanded on the limitations in the discussion. We moved details about the recruitment process from the supplement to the main manuscript as requested.
3. Focus groups. We agree with R1 that information about the focus groups were included sparingly in the original manuscript. This is partially due to the nature of the manuscript being a part of a much larger project; as such, the focus groups were designed to answer many research questions that go beyond just designing a peacebuilding scale. Based on R1’s feedback, we have revised our manuscript to more explicitly reflect that and only focus on portions of the focus group study (questions, analysis process, and answers) that speak directly to the scale development process. We have added some example quotes and connected them to the scale items.
4. Three types of peacebuilding. We have clarified the three levels in the introduction. We expand here: we are rooted in the sociological ecological model in developmental psychology (Bronfenbrenner) and peace studies (Lederach), who use those respective labels for each level. Although the DPM levels (microsystem/relational change, mesosystem/structural change, macrosystem/cultural change) share some conceptual and even terminological overlap with Galtung’s framework (direct, structural, and cultural violence), they are derived from different theoretical traditions and are not intended as direct equivalents. We recognise that there are differences in terminology across

disciplines and subfields, but we hope that the description provided adds clarity. We further acknowledged the terminology distinction (i.e. Galtung’s structural and cultural violence) in the introduction/discussion.

5. Presentation of statistical results. We agree that there was a lack of detail about the process and outcomes of freeing intercepts in our test of measurement invariance. Our aim was to highlight that, despite how different these contexts were, most subscales only required a small number of intercepts freed to achieve measurement invariance. We interpret this in more detail in both our results and discussion now, and also list recommendations about the application of this scale to other contexts given these findings.
6. Structure. We have revised the structure of our manuscript so methods precede results.

Reviewer 2 (R2)

We thank R2 for recognizing the work and effort in this manuscript, as well as the importance of the topic. As pointed out by both Editor and R1, the structure of the manuscript has been addressed.

In addition, we have addressed the following methodological concerns:

1. Mixed-methods design. We now explicitly state we are using a sequential mixed-methods design (page 2), and more details have been added to Figure 1; we also revised the section headers so readers can more easily connect the graphic to the corresponding sections. In addition, more details about the methods have been moved from the supplement to the main manuscript.
2. More information about Phase 1 & Qualitative Data Analysis. Details regarding the literature, recruitment, procedure, and analysis (portions relevant to scale development;) were added and/or transferred from the supplement (pages 3-4; see also (see R1.3, R2.2). Citations of previous literature that informed the focus group questions were added.
3. Qualitative Data Analysis. We now expand on the procedure for generating scale items from the focus group transcripts (see R1.3, R2.2), describing relevant steps and citing previous literature in more detail (page 3-4).
4. Procedure for initial reduction of items. We revised the manuscript to focus on only portions of the study relevant to the research questions for this paper (rather than the larger project, designed to answer multiple research questions over multiple papers). We clarify which actors were involved in each step of the process. See the supplement (S7) for a table labelling each level of the items from the initial item pool.
5. EFA process. We agree with R2 that the emergence of a four subscale structure was surprising. However, retaining a larger number of items from the macrosystem level was beneficial, as this was understudied in the peacebuilding literature. Furthermore, although we have an initial framework (the DPM), an EFA was selected precisely because it allows for modifications (and/or additions to) guided by data. We more explicitly outline the steps for item reduction in the main manuscript. However, our understanding is that EFA can be applied using both item loading and theoretical fit (e.g., Flora & Flake 2016; Lloret-Segura et al., 2014; Tabachnick & Fidell, 2001; Ziegler; 2014; Henson and Roberts, 2006), and would appreciate additional guidance if this is not satisfactory. Although we do not have the sufficient sample size to conduct a split half EFA (from just the Northern Irish sample), we have updated the figures to include

factor loadings (Figures 2-5) and revised to include alpha coefficients along with McDonald's omega (page 10). Following R3.5's recommendation to report McDonald's omega, there appears to be stronger support for the reliability for the mesosystem subscale; nevertheless, we acknowledge this limitation in the discussion.

6. Two items in Israel. We omitted two items due to recommendations from our collaborators (now expanded on page 5), due to the sensitive nature of this context. Specifically, intergroup contact (item 1) is not present in Israel due to ongoing tensions, and protest art (item 2) is extremely uncommon. We articulate additional implications for applying the scales in the discussion.
7. Account of CFA. Northern Ireland did indeed serve as the baseline model used for comparison, and the rationale behind this was outlined on page 2 and now further explained on page 6. We agree that the original CFA figures could be improved, and have now added separate figures for all contexts.
8. Measurement Invariance. Outcome results are fully listed in Table S32-S43 in the supplement. We also now report more results explicitly in the main manuscript (including the number of intercepts freed and discussing the lack of consistent patterns in across and within case comparisons), but still opt to keep most of the details in the supplement due to word count and space limitations. Of course, we are open to suggestions on how to best represent supplemental findings succinctly in the main manuscript.
9. Test of validity. We agree that the steps for data reduction were not outlined clearly in the original manuscript. We have revised the manuscript to clarify the procedure related to item reduction and omission (see R2.4, R2.5, R2.6).
10. Scale validation rooted in DPM. We agree the original phrasing of this sentence was confusing and have rephrased it (page 18).
11. Tense. We have edited the manuscript to ensure verb tense is consistent.

Reviewer 3 (R3)

We thank R3 for acknowledging the strength in our manuscript for its cross-cultural nature and difficult to recruit samples.

We address the following concerns:

1. Structure. See R1.6.
2. Lack of information in main manuscript. Content has been moved from supplement (including sample demographics, details about EFA, likert scale points, recruitment sources, item reduction process, and information about translations) (see also R1.3, R2.2, R2.3, R2.4).
3. EFA criteria. Results from parallel analysis have been added.
4. Tests of dimensionality. We clarified that measurement invariance was conducted pairwise, with Northern Ireland as the reference group. We agree with the reviewer that the third item does not necessarily fit under voting, but rather youth engagement in formal political processes; we have renamed this subscale to Voting/Politics. All EFA and CFA correlations are reported and discussed (see updated Figures 2-5). Each subgroup used to test measurement invariance has been clarified. Finally, more details have been added on partial variance testing in both the results and discussion sections (implications of how this scale should be applied given these invariances) (see R2.8).

5. Tests of reliability. We clarified that measures of reliability were calculated across all sample and demographic groups, and added reporting of omega (see R2.5), as well as reporting additional results by case (in the Supplement).
6. Tests of validity. We agree that the distinction between convergent and discriminant validity should be clarified. More details have been provided about the exact measures (moved from the methods section at the end). We have also added a paragraph (page 5) on how each construct would relate to peacebuilding in general and why they were selected. Furthermore, we clarify that we are not addressing directionality in our tests of convergent validity. We have revised the language to remove potential misunderstandings.
7. Figures and Tables. Notes were added to Figure 6 (originally Figure 1) to describe demographic controls and latent vs. measured variables; notes were added to Table 2 to describe discriminant validity.
8. Cross-case analysis. We now explicitly note this as exploratory (page 13-14) and clarify the motivation behind reporting (i.e., transparency). Statistical tests were specified and additional results (mean differences, SE) were reported.
9. Minor revisions. All minor revisions have been addressed.

We sincerely thank the editor and reviewers for your thoughtful feedback, which has helped to strengthen the manuscript.

We are happy to address any further comments or clarifications as needed.

Dear Reviewers,

We sincerely thank the reviewers for their thoughtful comments and feedback. We have carefully addressed all points raised, and all major revisions and sections relevant to reviewer comments have been marked in blue in the updated manuscript. Below, we provide a detailed point-by-point response.

Review #1:

1. Characteristics of respondents: This information has been added to the main text (lines 137-144).
2. Macrosystem beliefs: We thank the reviewer for their thoughtful insights on these distinctions. We agree that alternative frameworks can help contextualize the current findings, but note that (1) our measure captures the norms and cultural beliefs about intergroup contact, not an individual's own intergroup contact *behaviour*, and (2) we have added a brief section in the discussion that suggests future research to address two key distinctions: beliefs (the YPBS) vs. behaviour, as well as institutional vs. non-institutional participation.
3. Across- and Within-Case Comparisons: A brief section has been added to the discussion to reflect on gender and age comparisons (lines 582-587), but we also highlight sections in the manuscript that outline why we may want to be cautious in drawing strong conclusions from these differences (lines 472-475; 592-595). Similarly, the cases were selected due to their level and recency of conflict (see lines 97-114 for justification); we note that information about the type of conflict can be found in the Supplement (Note S1: Conflict History Overview).
4. Typos have been addressed with the review by an external reader, familiar with APA, but not an author on this paper.

Reviewer #2:

1. Typos and tense issues have been addressed through review by an external reader.
2. We clarify that the qualitative research (Study 1) was conducted prior to October 7th 2023, but the quantitative data (Study 2) was collected approximately half a year after October 7th. In Israel, protest art was not supported by our qualitative research; that is, despite active prompting as part of the focus group guide, youth did not endorse protest art as a viable or sensible form of peacebuilding. On that basis, and to keep the Israeli survey aligned with themes that participants raised as central, we did not include the protest-art item in Israel. Agreeing with R2, we have revised the manuscript with this explanation.
3. We have rephrased the methods so we more clearly refer to them as "Youth Peacebuilding Items" (under Measures) rather than "Scale" to avoid confusion with the finalized YPBS.
4. Table S7 has been revised. To further clarify the selection process, a brief summary of the rationales used for each item reduction has been added to Table S7. In addition, we reduced the number of tables (especially for EFAs) to include full reporting only for models we considered in the process, which is now summarized in Tables S9, S12, and S15.
5. See R2.1.

Reviewer #3:

We thank reviewer 3 for noting the careful revision, and the improved structure and flow thanks for the first round of expert feedback.

We greatly appreciate the reviewers' constructive suggestions. We believe that the manuscript has been substantially improved as a result of this feedback, and we hope that the revisions address all concerns raised.

Sincerely,

Vivian Liu